# Sorption of ^137^Cs and ^60^Co on Titanium Oxide Films in Light Water Reactor Primary Circuit Environment

**DOI:** 10.3390/ma15124261

**Published:** 2022-06-16

**Authors:** Nikita A. Glukhoedov, Vitaliy N. Epimakhov, Sergey N. Orlov, Anastasiya A. Tsapko, Aleksandr A. Zmitrodan, Grigoriy A. Zmitrodan, Mikhail Yu. Skripkin

**Affiliations:** 1Institute of Chemistry, Saint-Petersburg State University, 7/9 Universitetskaya Emb., 199034 St. Petersburg, Russia; n-glukhoyedov@rambler.ru (N.A.G.); orlov.s.n.1989@yandex.ru (S.N.O.); 2Federal State Unitary Enterprise “Alexandrov Research Institute of Technology”, 72, Koporskoe Shosse, 188540 Sosnovy Bor, Russia; evn@niti.ru (V.N.E.); tsapko.aa@niti.ru (A.A.T.); zaa72@mail.ru (A.A.Z.); zmitrodan.ga@niti.ru (G.A.Z.); 3Institute of Nuclear Industry, Peter the Great St. Petersburg Polytechnic University (SPbSU), 29, Polytechnischeskaya, 195251 St. Petersburg, Russia

**Keywords:** light water nuclear reactor, titanium alloys, sorption, long-lived radionuclides

## Abstract

This paper discusses the processes of the long-lived ^137^Cs and ^60^Co immobilization on titanium surfaces in simulated light water reactor primary circuit environments. This study is prompted by numerous problems in both the maintenance of equipment during reactor operation and the dismantling of the reactor after the completion of the operation, which is associated with contamination of working surfaces with long-lived radionuclides. The composition of the oxide films formed on the surface of commercial titanium alloy ПT-3B has been studied with specimens prepared in autoclave test conditions and surface samples from the pipeline sections to which the primary coolant was applied. These films on the coolant pipeline surface consist of a titanium dioxide layer tightly adhered to the pipeline metal surface and weakly fixed deposits—crystallites comprised of titanium oxides and other corrosion products (oxides and hydrated oxides of iron, nickel, chromium etc.). The radionuclide composition of the samples was studied by gamma-spectrometry. It is shown that the mechanism of titanium-surface contamination with ^137^Cs is by physisorption, contamination level increases upon the presence of dispersed particles. For ^60^Co, both sorption and deposition onto surfaces are observed.

## 1. Introduction

Titanium alloys are widely used in the nuclear industry as structural materials. Titanium dioxide anatase or rutile films are formed on titanium alloy surfaces when they come into contact with the primary coolant water. The film surface also includes several crystallites of titanium dioxide or ilmenite (FeTiO_3_) [1]. According to data of Buddington [2] the formation of several other mixed oxides of iron and titanium, namely ulvospinel Fe_2_TiO_4_ (Fe^+2^) and pseudobrookite Fe_2_TiO_5_ (Fe^+3^), is possible with different oxidation states of iron atoms. Under these conditions of the nuclear reactor primary circuit environment with super-critical parameters of coolant water crystallites containing chromium, zirconium and molibdenum can also form [3]. 

Co-deposition of iron- and titanium-containing corrosion products onto the surface would result in contamination of the film by long-lived radionuclides—activated corrosion products, such as ^60^Co (5.27 years half-life). Cobalt can be immobilized on the alloy surface by the co-deposition mechanism and also through ion exchange: hydrated titanium dioxide’s ability to sorb mono- and divalent cations with the formation of inner-sphere surface complexes is well known from the literature [4,5]:~Ti-OH + M^n+^ → ~Ti-OM^(n−1)+^ + H^+^(1)

The ion-exchange mechanism enables immobilization on the titanium surface of radionuclides that are found mostly in the primary coolant water. As an example, the immobilization of long-lived ^137^Cs (30.17 years half-life) produced by nuclear fission of uranium-235 can be considered. As is shown by Doshi [6], ^137^Cs is effectively adsorbed from water onto hydrated titanium dioxide: the distribution coefficient between the liquid and solid phases is 178. Venkataramani [7] has found that sorption of ^137^ Cs on hydrated titanium dioxide becomes noticeable at pH 4 and increases significantly upon transfer to alkaline medium—at pH 10 sorption is more than 10 times higher than at pH 4. Metwally [8] studied thermodynamic parameters of caesium ion sorption onto titanium dioxide. According to his data, upon the increase of temperature Gibbs free energy of the sorption process decreases, which makes this process more favorable from a thermodynamic point of view—reaction ability of titanium dioxide increases, and after the temperature rises from 25 to 50 °C, the amount of caesium sorped approximately increases twofold. At present, the effective sorbents based on titanium dioxide to remove caesium and cobalt radionuclides from aqueous solutions are under development. Silica gel [9], poly ethylene glycol [10], are potassium-nickel hexacyanoferrate [11] are considered as composite bases for these sorbents.

Radionuclides ^137^Cs and ^60^Co are characterized by a long half-life and significant energy of gamma-rays (661.6 and 1173.2; 1332.5 keV) that determines their radioecological hazard and complicates the disablement of nuclear-industry objects [12,13]. Moreover, ^137^Cs and ^60^Co are the typical representatives of two common kinds of radionuclides—volatile products of uranium fission and activated corrosion products, respectively. These factors determine the choice of ^137^Cs and ^60^Co as the subjects of this study. Contamination of titanium alloy surfaces with long-lived radionuclides causes problems in both the maintenance of equipment during reactor operation and the dismantling of the reactor after the completion of the operation. Moreover, the information about the mechanism of their interaction with alloy surfaces, as well as about their contamination level, is necessary for the development of effective methods for their removal.

Currently, the information about the mechanism of radionuclide interaction with oxide films of light water nuclear reactors is scarce and insufficient. Kasahara [3] has studied the composition and structure of oxide film corresponding to the primary circuit of a newly developed super-critical reactor (coolant temperature 550 °C, pressure 250 atm) that significantly differs from the conditions realized in the majority of existing reactors (temperature 200–300 °C, pressure up to 150 atm). Bignon [1] only studied the composition of film and not its contamination with radionuclides. Some authors [6,7,8,9,10,11] studied sorption of cobalt and caesium radionuclides but either freshly precipitated hydrated titanium dioxide or nanocomposites from hydrated and non-hydrated titanium dioxide were used as sorbents. At the same time, freshly precipitated hydrated titanium dioxide has a large specific surface area (ca. 200 m^2^/g [14,15]) and a significant amount of hydroxyl groups on its surface (2–5 nm^−2^ [8]). As a result, the level of radioactive contamination and factors determined its values for hydrated titanium dioxide, and the oxide film on titanium alloys should differ significantly.

In connection with the above mentioned, it appears important to carry out experimental research for measuring the level of titanium alloy surface contamination in the nuclear reactor primary system with the long-lived ^137^Cs and ^60^Co, determining the conditions that influence this level, and understanding the mechanism of the radionuclide immobilization onto the alloy oxide film surface. 

## 2. Materials and Methods

Experiments were performed with commercial ПT-3B alloy. This alloy along with ПT-7M alloy is the most widely used in nuclear industry [7]. The ПT-3B alloy composition is as follows: Ti 91.4–95.0%; Al 3.5–5.0%; V 1.2–1.5%; Zr up to 0.3%; Fe up to 0.25% by mass.

For the experiments 2.0 N standardized solution of nitric acid, 27% *w*/*w* aqueous hydrogen peroxide, 99.6% titanium(IV) chloride, 28% ammonium hydroxide, solid oxalic acid dihydate (98%), iron(III) nitrate nonahydrate (98+%), potassium dichromate (99%), and nickel(II) nitrate hexahydrate (98%) were used. These chemicals were supplied by Alfa Aesar. The treatment solutions were prepared by dilution. Standard aqueous solutions of ^137^Cs and ^60^Co were supplied by Isotope JSC, Russia.

The radionuclide content of surface deposits on the pipelines made of ПT-3B alloy and contacted with the light water reactor coolant water was determined. Model autoclave experiments with specimens of this alloy were also carried out. 

The composition of weakly fixed deposits was analyzed using wipe samples from the pipeline surface. Sampling was performed by pressing swabs of lint-free cloth (madapollam) and PU foam soaked in ethanol or nitric-acid solution (1 mol/kg concentration, prepared from 2.0 N Standardized Solution (Alfa Aesar) by weighting) onto the surfaces of interest. The wipe samples were taken from linear parts of the 90 mm and 150 mm diameter pipelines and at the pump discharge pipeline. 

Segments of the 14 mm diameter pipeline were examined to determine the total level of the alloy surface contamination. The pipeline was cut into two parts and the inside surface that came into contact with water was inspected. 

The autoclave experiments were carried out on ПT-3B titanium alloy specimens with the mean roughness depth R_Z_ = (0.8–1.15) µm. The specimens were freed from the original oxide film by putting them in an aqueous solution of oxalic acid and hydrogen peroxide (concentration of each agent 5 g/kg) for 24 h. The solution pH was adjusted to specified values of 4, 7, or 10 by using ammonia or oxalic acid. The specific activity of each radionuclide in the solution was about 1 × 10^6^ Bq/kg. The solution volume in the autoclave was 60 mL. The autoclaves were made of 08X18H10T steel. The autoclaves were heated at 200 °C for 40 h. In several experiments to study the kinetics of deposits’ formation, heating was interrupted after 3, 5, 10 and 20 h. After, measurement samples were returned to autoclaves to continue heating. The design of the autoclaves used is presented at Figure 1.

An aqueous solution without carrier phase was used as ^137^Cs source. A standard solution of cobalt chloride in hydrochloric acid was used as ^60^Co source. The stable cobalt nuclide concentration in the autoclave solutions accounting for dilution (owing to the carrier phase presence in the standard solution) was not larger than 1–5 µg/kg. 

In the experiments with the addition of a carrier phase (with the same composition as that of the corrosion products of structural materials), saline solutions of titanium (IV) (chloride), iron (III) (nitrate), chrome (potassium bichromate), and nickel (II) (nitrate) were used. The concentration of each metal in the solution was 5 mg/kg.

After the completion of each experiment, the alloy specimens were taken from the autoclave and the remainder of the solution was thoroughly removed from their surface by filtering paper. Moreover, 40 mL liquid samples were taken from the autoclave after each experiment for measuring the radionuclide activity in the liquid phase.

For the real samples taken from pipeline of nuclear reactor (wipes, crystallites, etc.), the unique results are presented. For the autoclave experiments, the averaged data obtained for three samples are presented; the margin of error does not exceed 10%.

The radionuclide composition in the specimens and solution was determined by gamma spectrometry with the “Гaммa-1П” gamma ray spectrometer with semiconductor detector. 

The chemical composition in the aqueous solutions was determined by atomic emission spectroscopy with the iCAP-6500 inductively coupled plasma optical emission spectrometer.

The morphology of the oxide film on the alloy specimens was analyzed in the NovaNanoSEM 450 field-emission scanning electron microscope. The composition of individual particles on the film surface was also determined in this instrument by x-ray microanalysis using the Quantax 400 accessory. 

Based on the data of the radionuclide and elemental composition of the specimens and their surface areas (or the alloy surface areas from which wipe samples were taken), the total level of surface contamination (for the alloy specimens) and the level of weakly fixed (removed by wiping) contamination were calculated. 

## 3. Results

### 3.1. Wipe Samples

Figure 2 compares ^60^Co and ^137^Cs activities in acid- and ethanol-soaked wipe samples taken from two pipeline sections.

The activity of ^60^Co and ^137^Cs is in the range of 150–220 Bq and 50–80 Bq, respectively. The activity levels almost do not differ between the acid- and ethanol-soaked wipe samples.

Figure 3 presents data on the level of weakly fixed contamination on the titanium alloy surface. The presented values are calculated from analysis of the wipe samples and averaged over the pipeline sections where these samples were taken. 

The ^137^Cs contamination of the examined pipeline surfaces is in the range of 20–50 kBq/m^2^. The level of ^60^Co contamination on the pipeline surfaces is very different from that of ^137^Cs: 6–8 kBq/m^2^ for long pipe sections with laminar coolant flow (pipelines of 90 mm and 150 mm diameter) and about 130 kBq/m^2^ for the pump discharge pipeline where turbulence occurs).

### 3.2. Pipe Specimens

Figure 4 presents the image of the pipeline inside surface (a) and its photo taken with microscope (b).

As shown in the figure, the inside surface of the pipe specimen that was in contact with the coolant water is coated by a solid light-yellow oxide film. Some surface roughness is visible. However, no apparent porosity and developed surface is observed.

Individual crystallites are also seen on the oxide film. The elemental composition (without oxygen) of these crystallites is presented in Table 1. The images of two selected crystallites are presented in Figure 5.

The crystallites are mostly composed of titanium compounds. They also contain carbon and steel corrosion products—chrome, nickel, and iron. The size of the crystallites ranges from 0.8 to 3 µm.

Figure 6 shows the level of ^60^Co and ^137^Cs contamination on the surface of four pipeline sections. 

As shown in the figure, the contamination is fairly uniformly distributed along the pipeline length, without local activity peaks being seen. The total surface contamination level obtained from the examination results is almost identical to the level of weakly fixed contamination on the titanium alloy surface which was calculated from analysis of the wipe samples taken from the test circuit.

### 3.3. Autoclave Experiments

Figure 7 shows the level of ^60^Co and ^137^Cs contamination on the surfaces of ПT-3B alloy specimens after the autoclave experiments. 

As seen in Figure 7, the ^60^Co and ^137^Cs contamination on the surfaces of the specimens after the autoclave experiments using solution and without a carrier phase is at the same level as the contamination of the alloy surface in the test circuit. The solution pH has little if any influence on the contamination level of the specimens. The activity level of the radionuclides in the solution changed by less than 10% after the experiments, and no noticeable radionuclide deposition on the autoclave surface was observed. Additional experiments have demonstrated that contamination on the specimen surfaces develops during the first 3 h of heating and then remains almost unchanged. 

The radioactive contamination on the surface of the alloy specimens after autoclave experiments with a carrier phase (metal salts) at pH 7 reached 120 and 3100 kBq/m^2^ for ^137^Cs and ^60^Co, respectively. These values are 2 (^137^Cs) and 60 times (^60^Co) higher compared to other autoclave experiments without a carrier phase. After the autoclave experiments, the radionuclide activity level in the solution decreased by 3–4 times for ^137^Cs and more than 10 times for ^60^Co.

## 4. Discussion

The oxide film on the coolant pipeline surface consists of a titanium-dioxide layer tightly adhered to the pipeline metal surface and weakly fixed deposits—crystallites comprised of mixed titanium oxides and alloy-corrosion products. The structure of the oxide film on the specimens taken from the coolant circuit is identical to that of the deposits on the specimens prepared in autoclave experiments reported in [1].

A wide range of the values of iron and titanium mass fraction in the crystallites suggests that combined particles of ilmenite, titanium dioxide, and magnetite in different proportion can form in the primary circuit environment and may also contain chrome oxide (III) and to some extent nickel oxide (II).

The level of surface contaminatiuon of pipelines by ^137^Cs calculated from the wipe analysis is between 20 and 50 kBq/m^2^, whereas the value obtained directly from the measurements of pipe samples is 40–80 kBq/m^2^. On the other hand, the corresponding radionuclide ^60^Co values are 6–8 and 10–20 kBq/m^2^ respectively. The only exception was observed for the wipe sample taken from the pipe-discharge pipeline—the ^60^Co contamination level reaches 130 kBq/m^2^, and the reasons are discussed below. In general, the comparison of the contamination of pipe samples to the contamination seen in the wipe experiments allows us to conclude that approximately 50% of contamination by ^137^Cs and ^60^Co radionuclides is associated with weakly fixed deposits.

Good agreement in the values between the total level of contamination on the pipe specimen surface and the surface contamination calculated from analysis of the wipe samples indicates that ^137^Cs and ^60^Co in the primary circuit conditions are mainly associated with weakly fixed deposits (crystallites) or adsorbed on the titanium alloy surface. This conclusion is confirmed by the measurement of activity levels in the acid- and ethanol-soaked wipe samples taken from the pipeline surface: the radionuclide content of these samples is almost identical, even though acid solutions dissolve tightly adhered oxide deposits more efficiently than ethanol does. 

The ^137^Cs contamination on the titanium alloy surface is in a fairly narrow range for both pipeline surfaces and specimens prepared in the autoclave experiments. In almost all autoclave experiment conditions the mechanism of contamination was sorption but not deposition (change of the solution radioactivity level during the experiment was not as noticeable as that observed in case of deposition). Therefore, it can be inferred that ^137^Cs contamination on the pipeline surface was also determined by the surface sorption capacity. The only exception observed in the present work was the case of a carrier phase deposition from solution: in that case the ^137^Cs activity in the solution significantly decreased, while the level of ^137^Cs contamination on the titanium alloy surface increased by about two times. This behavior is probably explained by co-deposition of ^137^Cs and particles composed of metal oxides—steel corrosion products. 

The level of ^60^Co contamination on the pipelines in the test circuit is different from that of ^137^Cs. Note a high level of ^60^Co contamination on the pump discharge pipeline where turbulent flow is observed. This fact indicates that contamination forms by the deposition mechanism. Almost complete transfer of ^60^Co onto the surface is also observed in the case of carrier phase deposition from solution. In this case, the level of ^60^Co contamination on the alloy surface is hundreds of times larger than the level of ^60^Co sorption on titanium. The nearly complete transfer of ^60^Co radionuclide from solution to surface of titanium alloy is also observed in presence of a carrier phase. Under these conditions, the level of surface contamination with ^60^Co (3100 kBq/m^2^) is a hundred times higher compared to those of titanium alloy under technological conditions (6–20 kBq/m^2^) or under autoclave conditions (25–50 kBq/m^2^) without a carrier phase.

As known from the literature, the value of cesium and cobalt sorption onto hydrated titanium dioxide through ion exchange strongly depends on solution pH. Sorption becomes noticeable at pH = 4 and significantly increases after transition to basic conditions. This may be due to the electrostatic interaction between ions and titanium surface [5]. Since there is no distinct correlation between the level of ^60^Co and ^137^Cs contamination on the specimen surfaces and the solution pH in the autoclave experiments, we suggest the domination of radionuclide physisorption onto ПT-3B titanium-alloy surface in the light water reactor primary circuit environment. 

## 5. Conclusions


The titanium-alloy surface under simulated conditions of the light water reactor primary circuit is coated with a solid titanium dioxide film. Mixed oxides of titanium and steel corrosion products are loosely deposited on this film. The ^137^Cs and ^60^Co radionuclides are adsorbed onto the oxide film surface and loose deposits.In the primary circuit water-chemistry environment, contamination of the titanium alloy surface with ^137^Cs is due to physisorption. The level of contamination depends on the surface sorption capacity and is about 20–60 kBq/m^2^ for ПT-3B alloy. This value can increase by 2–3 times due to co-deposition of corrosion product particles when they are present in large amounts in the coolant. The level of ^60^Co contamination on the titanium-alloy surface is primarily determined by the concentration of this radionuclide and other corrosion products in the coolant and the contamination mechanism is through deposition.


**Future work**. The study of solution composition effects on the composition and structure of oxide films and crystallites on the surface of titanium alloy will be the key question in the next phase of this work.

## Figures and Tables

**Figure 1 materials-15-04261-f001:**
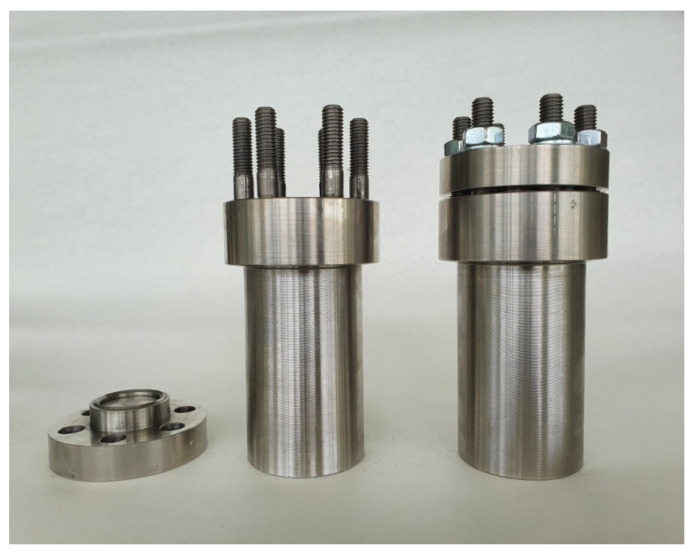
Autoclaves used in the experiment.

**Figure 2 materials-15-04261-f002:**
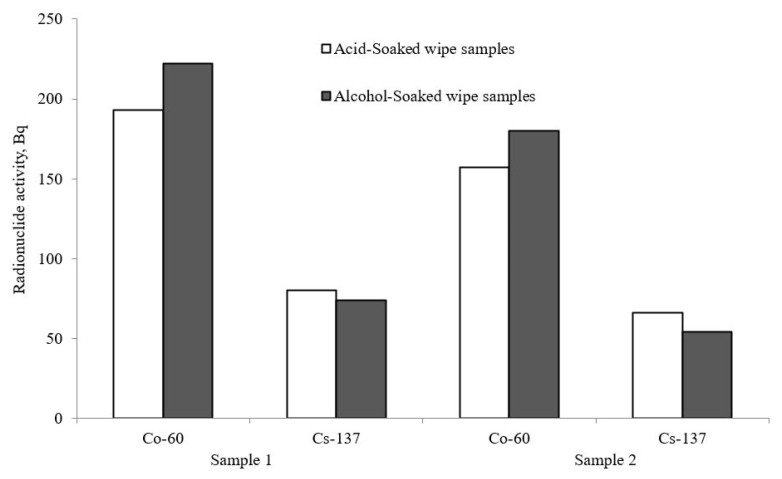
Comparison of radionuclide activity in ethanol- and acid-soaked wipe samples taken from the surface of linear parts of the coolant pipelines.

**Figure 3 materials-15-04261-f003:**
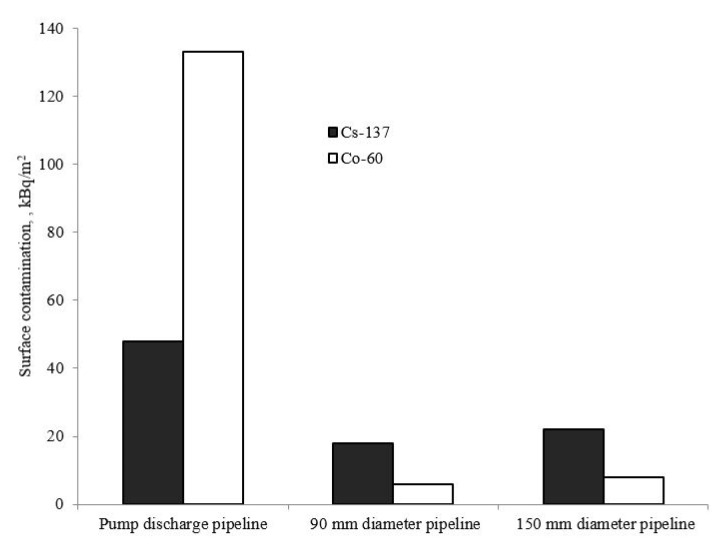
Weakly fixed radioactive contamination on the surface of the pipelines in the test circuit.

**Figure 4 materials-15-04261-f004:**
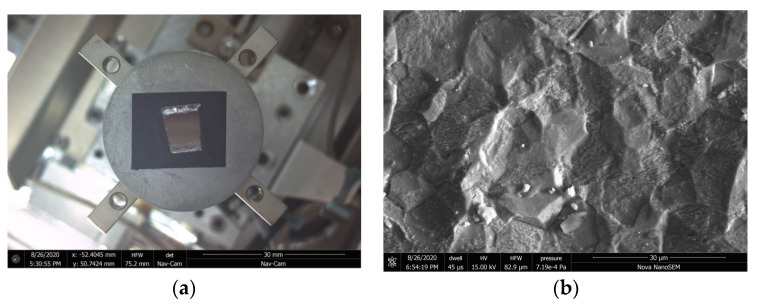
Image (**a**) and morphology (**b**) of the specimen of 14 mm diameter pipeline.

**Figure 5 materials-15-04261-f005:**
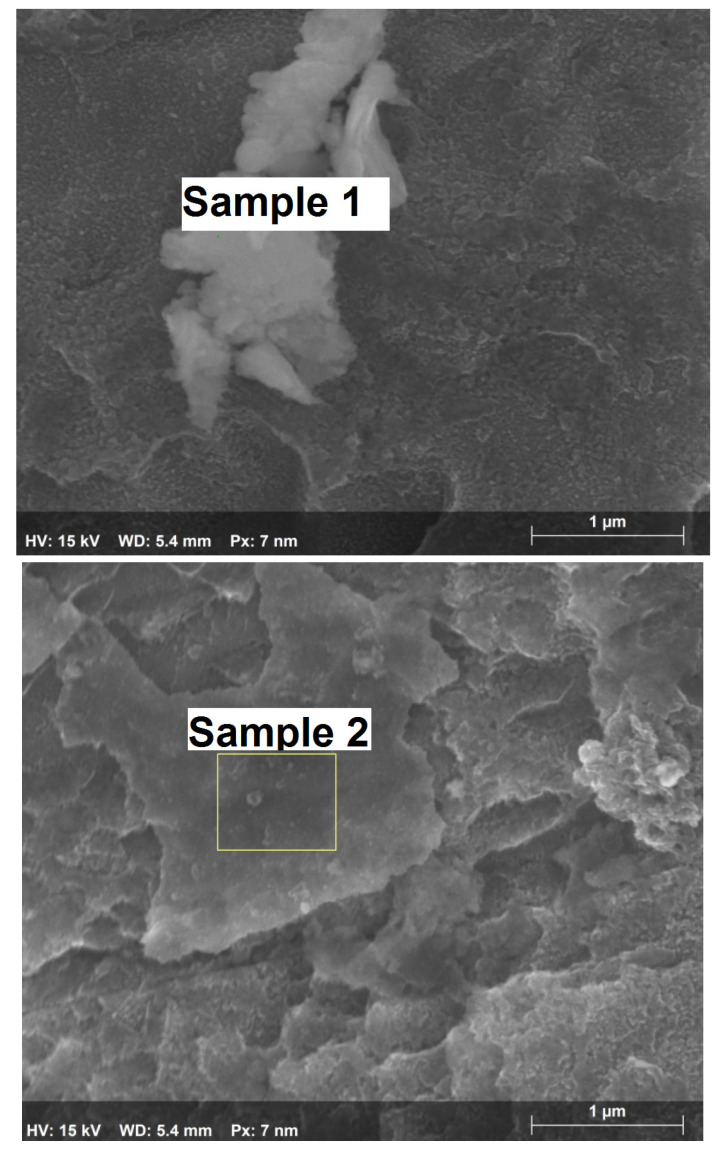
Images of the selected crystallites (sample numbering corresponds to that in Table 1).

**Figure 6 materials-15-04261-f006:**
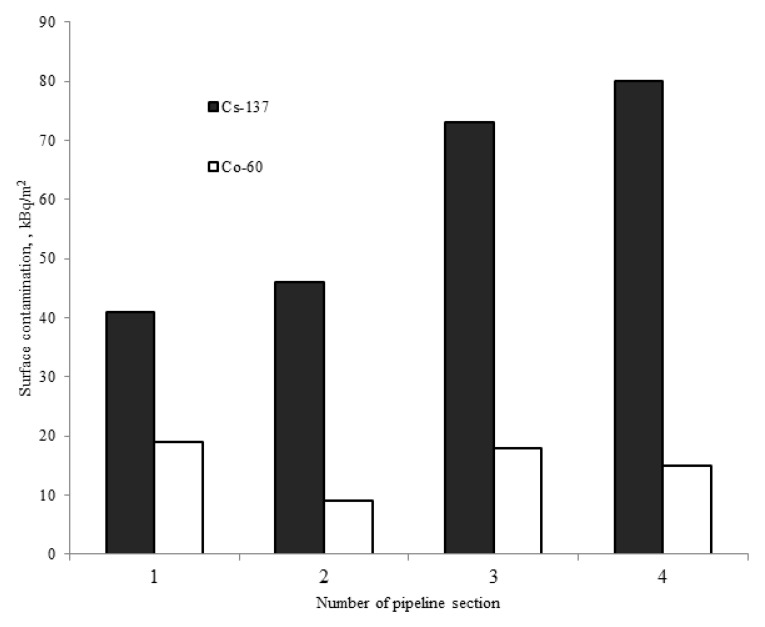
Radioactive contamination on the surface of the coolant pipeline specimen.

**Figure 7 materials-15-04261-f007:**
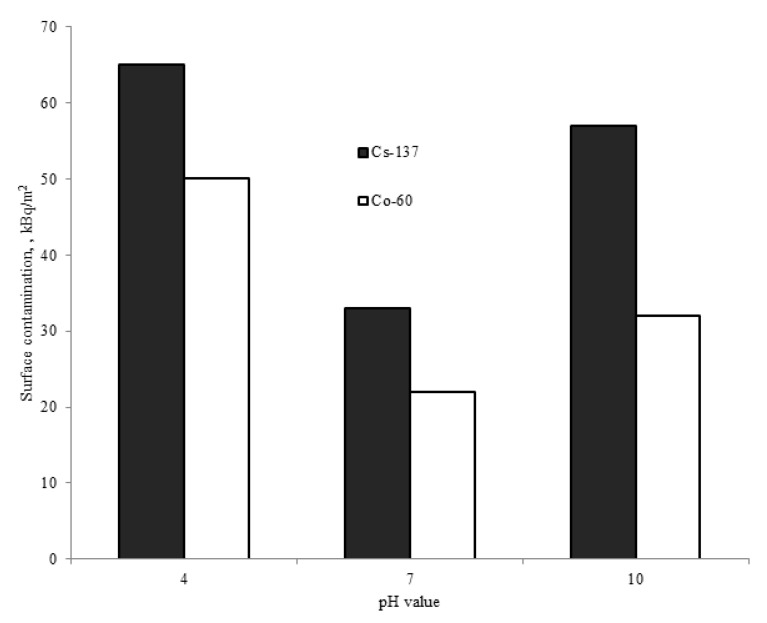
Radioactive contamination on the surface of the alloy specimens after autoclave experiments.

**Table 1 materials-15-04261-t001:** Elemental composition of crystallites.

Sample Number	Element Mass Fraction, %				
	Ti	Fe	Ni	Cr	C
1	63	17	-	1	18
2	94	1	-	-	5
3	73	6	1	10	10
4	79	4	1	8	8
5	95	-	-	-	4

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
