# Peer review of "Sorption of 137Cs and 60Co on Titanium Oxide Films in Light Water Reactor Primary Circuit Environment"

_materials, 2022, doi:10.3390/ma15124261_

Round 1
Reviewer 1 Report
Comments on materials-1722367
After a careful peer-reviewing process, I must inform you that, the subject of this paper is interesting and can be considered for publication in Materials after a MAJOR REVISION. I believe that the paper contains relevant information for the scientific community, especially in the recent advances in Ti and its alloys are used as structural materials. I believe that the results are informative but must be well organized and improved in the next revision(s). Therefore, there are some questions about this submission and some revisions are necessary for this work. The major/minor issues are indicated as follows:
1- The quality of images 1 and 2 is too low. Please improve the quality of these figures. Also the sclebars of some figures also missed (e.g., Fig. 3 a and b).
2- The abstract is not well written. Some details of the methodology are missed in the abstract. Please revise this section.
3- In this manuscript, results on wipe and pipe samples are well presented, but a discussion on the obtained results must be completely provided in this manuscript. As can be seen, a “comprehensive” and “comparative” discussion is missed in this work. This should be provided as well in section 4.
4- Please provide major findings in ‘Conclusions” section in a bullet-point style.
5- There is no description of the future plans for research in the first part of the “Conclusions” section. This should be completed in this section.
6- Recently published references are beneficial for this work. Please check and use new references focused on your work.
7- For the statement: "The level of 60Co contamination on the pipelines in the test circuit is different from 225 that of 137Cs. Note a high level of 60Co contamination on the pump discharge pipeline 226 where turbulent flow is observed. This fact indicates that contamination forms by the 227 deposition mechanism. Almost full transfer of 60Co onto the surface is also observed in 228 the case of carrier phase deposition from solution. In this case the level of 60Co 229 contamination on the alloy surface is hundreds of times larger than the level of 60Co 230 sorption on titanium." please provide reference for justification.
Author Response
First of all, we greatly appreciate the reviewer for very valuable comments that allow us to improve the manuscript. The detailed report of the changes introduced is given below.
1- The quality of images 1 and 2 is too low. Please improve the quality of these figures. Also the sclebars of some figures also missed (e.g., Fig. 3 a and b).
Figures 1 and 2 have been updated and scalebars were added at Fig. 3.
2- The abstract is not well written. Some details of the methodology are missed in the abstract. Please revise this section.
Abstract have been revised.
3- In this manuscript, results on wipe and pipe samples are well presented, but a discussion on the obtained results must be completely provided in this manuscript. As can be seen, a “comprehensive” and “comparative” discussion is missed in this work. This should be provided as well in section 4.
Some additional lines are added in Section 4.
4- Please provide major findings in ‘Conclusions” section in a bullet-point style.
The style of “Conclusions” has been changed according to the recommendation.
5- There is no description of the future plans for research in the first part of the “Conclusions” section. This should be completed in this section.
The description of future work has been added.
6- Recently published references are beneficial for this work. Please check and use new references focused on your work.
Several new references have been added
7- For the statement: "The level of 60Co contamination on the pipelines in the test circuit is different from 225 that of 137Cs. Note a high level of 60Co contamination on the pump discharge pipeline 226 where turbulent flow is observed. This fact indicates that contamination forms by the 227 deposition mechanism. Almost full transfer of 60Co onto the surface is also observed in 228 the case of carrier phase deposition from solution. In this case the level of 60Co 229 contamination on the alloy surface is hundreds of times larger than the level of 60Co 230 sorption on titanium." please provide reference for justification.
The text was modified to emphasize that we compare these data with another ones mentioned in this paper.
Reviewer 2 Report
In the present paper, the processes of the long-lived 137Cs and 60Co immobilization onto titanium surfaces in simulated light water reactor primary circuit environments are studied. The compositions of the oxide films on titanium alloy surfaces are analyzed. Additionally, the contamination mechanisms of titanium surfaces with the elements of 137Cs and 60Co are discussed. The results are interesting and useful in industry. However, several flaws should be improved.
- Why the content of Fe element in sample 1 is larger than that of other samples?
- As shown in Table 1, the elemental composition of crystallites for five samples can be found. However, the SEM pictures of sample 1 and 2 are analyzed in Fig. 4. The other SEM pictures are missed.
- The formats of table and figures should be further modified.
- The words and grammar in present study are needed to be promoted.
Author Response
First of all, we greatly appreciate the reviewer for very valuable comments that allow us to improve the manuscript. The detailed report of the changes introduced is given below.
- Why the content of Fe element in sample 1 is larger than that of other samples?
The difference in crystallite compositions can arise from the mechanism of their formation. That is the question for further studies.
- As shown in Table 1, the elemental composition of crystallites for five samples can be found. However, the SEM pictures of sample 1 and 2 are analyzed in Fig. 4. The other SEM pictures are missed.
As habitus of all the crystallites is very similar, the authors decided to show only some examples. If necessary, we can add the other ones but they do not provide important new information.
- The formats of table and figures should be further modified.
Figures have been updated.
Reviewer 3 Report
In this paper, the authors reported the Sorption of 137Cs and 60Co on titanium oxide films in light water reactor. I have the following comments on this paper
- Abstract: it is short, I suggest to the authors to add some results into this section (the main results)
- Introduction: the authors must add some previous works similar to this work and then show the novelty in this study
- Introduction: I found only 6 references in this section, and I think you must add more recent Refs (the introduction must contain 15 references in the average)
- Can you add one figure for the experimental work (in section 2)
- Please enhance the resolution of Fig.1 and Fig.2
- Why the authors focuses on these two radioisotopes: 60Co and 137Cs
- I think you can remove Table 2 and mention the information in the text since it contains only one information about the Surface contamination with radionuclide for 137Cs and 60Co
- Line 176: As is seen in this figure, ...I suggest to the authors to mention the number of figure instead of (As seen in this figure), you can say that As seen in Fig.6
- line 181: Additional experiments have demonstrated that contamination on the..........Do you mean additional experiments in this work, or other works? if other works you need to add some Refs
- The number of references in this work is only 8 and this is not enough, you need to add more Refs in the introduction, in section 2 and section 3
Author Response
First of all, we greatly appreciate the reviewer for very valuable comments that allow us to improve the manuscript. The detailed report of the changes introduced is given below.
- Abstract: it is short, I suggest to the authors to add some results into this section (the main results)
- Introduction: the authors must add some previous works similar to this work and then show the novelty in this study
- Introduction: I found only 6 references in this section, and I think you must add more recent Refs (the introduction must contain 15 references in the average)
Abstract and Introduction have been updated, some new lines and new references have been added.
- Can you add one figure for the experimental work (in section 2)
The figure is added.
- Please enhance the resolution of Fig.1 and Fig.2
Figures were updated.
- Why the authors focuses on these two radioisotopes: 60Co and 137Cs
Introduction is updated to emphasize the key role of these two isotopes.
- I think you can remove Table 2 and mention the information in the text since it contains only one information about the Surface contamination with radionuclide for 137Cs and 60Co
- Line 176: As is seen in this figure, ...I suggest to the authors to mention the number of figure instead of (As seen in this figure), you can say that As seen in Fig.6
Text is modified according to these comments.
- line 181: Additional experiments have demonstrated that contamination on the..........Do you mean additional experiments in this work, or other works? if other works you need to add some Refs
The description of these experiments is added in Section 2.
Reviewer 4 Report
Dear authors,
You did an interesting work but the presentation of it does not satisfy. My remarks are as follows:
Abstract is too short; it must be expanded. Please improve Abstract by adding the one or two sentences about the motivation for this work. Also, emphasize the importance of the studied problem as well as novelty. Further, list the used methods and key findings.
Introduction section does not provide sufficient background analysis and relevant references are missing. Namely, there should be more critical analysis of articles dealing with topic investigated in this manuscript.
P2 L56: Was the alloy produced in this investigation or is it commercial? How the chemical composition was obtained?
P3 L101-102: There is no need to optical emission spectrometry be in italic.
P3 L103-104: Please, describe in short the metallographic preparation of specimens used for FE-SEM analysis.
Please emphasize the method that was used for obtaining results in Table 1.
Were the images of samples 1 and 2 in Figure 4 taken at the same magnification? missing scale.
In Discussion, you should try to confront your obtained results with results from other researchers dealing with similar issue.
Please explain/emphasize a value of this research and its importance.
Please state/emphasize how many samples were investigated. Whether the numerical results in the figures 1,2,5 and 6 and tables 1 and 2 are average or refer to one sample each? If there were more samples/more measurements what are the standard deviations?
Best regards
Author Response
First of all, we greatly appreciate the reviewer for very valuable comments that allow us to improve the manuscript. The detailed report of the changes introduced is given below.
Abstract is too short; it must be expanded. Please improve Abstract by adding the one or two sentences about the motivation for this work. Also, emphasize the importance of the studied problem as well as novelty. Further, list the used methods and key findings.
Introduction section does not provide sufficient background analysis and relevant references are missing. Namely, there should be more critical analysis of articles dealing with topic investigated in this manuscript.
- Abstract have been updated, several additional lines have been added according to reviewer’s comments.
P2 L56: Was the alloy produced in this investigation or is it commercial? How the chemical composition was obtained?
- This is commercial alloy used in atomic industry. Its composition was provided by supplier.
P3 L101-102: There is no need to optical emission spectrometry be in italic.
P3 L103-104: Please, describe in short the metallographic preparation of specimens used for FE-SEM analysis.
- No special preparation has been made to avoid destroy of oxide film.
Please emphasize the method that was used for obtaining results in Table 1.
- X-ray microanalyses have been used.
Were the images of samples 1 and 2 in Figure 4 taken at the same magnification? missing scale.
- Scalebars are added
In Discussion, you should try to confront your obtained results with results from other researchers dealing with similar issue.
- Unfortunately, the majority of studies in this field were carried out at quite different conditions that does not allow to compare the results. Therefore the comparison is mostly with ref. [1] where the data obtained at similar conditions are presented.
Please explain/emphasize a value of this research and its importance.
- The corresponding changes were introduced into the text
Please state/emphasize how many samples were investigated. Whether the numerical results in the figures 1, 2, 5 and 6 and tables 1 and 2 are average or refer to one sample each? If there were more samples/more measurements what are the standard deviations?
- For real samples (Fig. 1, 2, 5, Table 1) from nuclear power circuit the unique data are presented because of the problem with the availability of a set of samples. As to the model autoclave experiments (Table 2, Fig. 6), the average data for three samples are given, esd does not exceed 10% that is comparable with standard error of measurements,
Round 2
Reviewer 1 Report
Accept
Author Response
Thank you a lot for your comments.
Reviewer 2 Report
In the present paper, the processes of the long-lived 137Cs and 60Co immobilization onto titanium surfaces in simulated light water reactor primary circuit environments are studied. The compositions of the oxide films on titanium alloy surfaces are analyzed. Additionally, the contamination mechanisms of titanium surfaces with the elements of 137Cs and 60Co are discussed. The results are interesting and useful in industry. The flaws have been revised.
Author Response
Thank you a lot for your comments.
Reviewer 3 Report
The authors revised the paper
Author Response
Thank you a lot for your comments.
Reviewer 4 Report
Dear authors,
thank you for accepted comments and their implementation in the manuscript.
Best regards
Author Response
Thank you a lot for your comments. English checking was made again.